



# Direct dating of overprinting fluid systems in the Martabe epithermal gold deposit using highly retentive alunite

Jack Muston[1], Marnie Forster[1], Conrad Alderton[2], Shawn Crispin[2], Gordon Lister[1]

[1]Structure Tectonics Team, Research School of Earth Sciences, Australian National University, Canberra, 2601 Australia
[2]PT Agincourt Resources, Martabe Mine, Sumatra, Indonesia

*Correspondence to*: Jack Muston (jemuston@gmail.com)

**Abstract.** The Martabe deposits in Sumatra, Indonesia formed in a shallow crustal epithermal environment (200-350°C) associated with mafic intrusions, usually recognised in domes, adjacent to an active right-lateral wrench system. Ten samples containing alunite were collected for high-resolution $^{40}$Ar/$^{39}$Ar geochronology, to determine if overprinting fluid systems

could be recognised. At the same time, ultra-high-vacuum (UHV) furnace step-heating $^{39}$Ar diffusion experiments were conducted, to determine the argon retentivity of the mineral grains being analysed. The heating schedule chosen to ensure Arrhenius data uniformly populated the inverse temperature axis, with sufficient detail to allow the application of the Fundamental Asymmetry Principle (FAP) during data analysis. The heating time for each step was chosen to ensure reasonable uniformity in terms of incremental percentage gas release during each step. Results show activation energies

between 360 – 500 kJ/mol, with normalised frequency factor between 1.89e14s$^{-1}$ and 8.62e18s$^{-1}$. Closure temperatures range from 390–519°C for a cooling rates of 20°C/Ma, giving confidence that the ages represent growth during periods of active fluid movement and alteration. The Martabe deposit formed at temperatures <200°C at a depth of < 2km. Five distinct alunite growth events can be recognised: (A) 3.48–3.46 Ma; (B) 3.24–3.22 Ma; (C) 2.51– 2.12 Ma; (D) 2.08–1.90 Ma; and (E) 1.70–1.40 Ma. Gold in the Purnama pit is the result of fluid rock interactions in periods C and D.

**1 Introduction**

Epithermal gold ± copper ± silver deposits form in shallow crustal environments (1-2 km depth). Low-sulphidation deposits form distal to their source magmas, through mixing and transport by deep groundwater fluids, and are characterised by a reduced sulphur species, H$_2$S. High-sulphidation deposits formed from magmatic fluids proximal to their source intrusion and are undiluted by groundwaters. In these low-pH conditions a suite of alteration minerals is formed (*e.g.*, dickite, alunite,

kaolin-dickite, pyrophyllite). The mineral alunite [KAl$_2$(SO$_4$)$_2$(OH)$_6$] is of particular interest as it is a potassium bearing mineral that has been shown to be a useful $^{40}$Ar/$^{39}$Ar geochronometer for dating alteration systems. It commonly forms in porphyry and epithermal gold systems when hot, highly acidic fluids interact and alter potassium-feldspars. In this acid-sulphate alteration, a subset of advanced argillic alteration is distinguished by the formation of alunite (Rye *et al.*, 1992).





In this research we used high precision 40Ar/39Ar geochronology to date 10 alunite samples from the Martabe epithermal
gold field in Sumatra, Indonesia (Fig. 1 and 2). The sample sites have been chosen to allow distinction of overprinting fluids
within the mineralised corridor. Previous work has dated alunite using 40Ar/39Ar geochronology and found that the
alteration system was active at between $3.30 \pm 0.2$ Ma and $2.00 \pm 0.2$ Ma (Sutopo, 2013). The Sutopo (2013) geochronology
was done on four samples (two of which were fusion ages). Our study will build on these results, increasing the number of
samples being analysed to provide a more detailed understanding of the alteration system at the Martabe gold field.

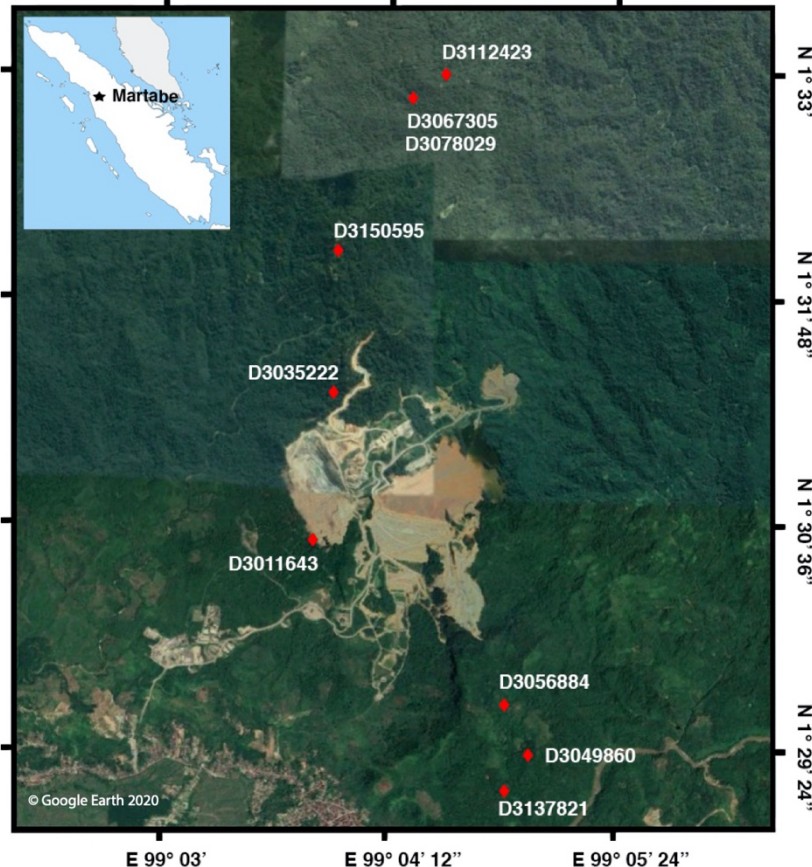


**Figure 1. Map of Martabe gold field, Sumatra, Indonesia. The location of the 10 alunite samples are shown with red diamonds.**

## 2 Geological Setting

Sumatra has a number of Neogene epithermal deposits of varying size and grade. The southern extent of the island has a
number of low-sulphidation epithermal deposits (*e.g.*, Mangani mine, Lebong district), while the north has predominately
high-sulphidation epithermal deposits (*e.g.*, Martabe mine, Meluak deposit, and Miwah mine). Over the last decade Sumatra
gold production has seen a significant increase, largely due to the successful development of the Martabe gold mine, which
began production in 2012.



The Martabe gold system was first discovered in 1997 after a positive result from geochemical surveys conducted on stream sediment within an exploration tenement. This was soon followed by field mapping, rock-chip sampling, and aeromagnetic

surveying, which led to the discovery of five deposits within a 7 km by 3 km corridor. The largest of these deposits is Purnama with a resource estimate of 4.3 million ounces of gold and 53 million ounces of silver (Agincourt, 2016). The host rock is a series of Tertiary mafic flow domes complexes and maar-diatreme breccias. There is a conjugated set of northeast extensional faults that are known to structurally control the fluid pathways for alteration and mineralisation within the Martabe district. These faults splay from the Sumatran Fault System, which is a 1600 km long wrench system that runs the

entire length of Sumatra (Levet, 2003). This right-lateral wrench system forms in response to the oblique subduction of the Indo-Australian plate (Barber and Crow, 2005).  Continual movement (in particular in conjunction with active volcanism) has provided long-lived fluid pathways, localising and enriching the deposits. We set out to explore the evolution of this fluid system using $^{40}Ar/^{39}Ar$ geochronology.





**Figure 2. (A) Sample PURNAMA P-01. Fine grained porphyritic volcanic andesite with possible hornblende, pervasive alunite-silica alteration, staining oxide (with disseminated hematite). (B) Sample D3011643. Contact between sediment and quartz vein. (C) Sample D3150595. Fine-grained clay samples with reddish-brown banding, alunite-clay altered. (D) Sample D3112423. Phreatomagmatic breccia, polymict, trace sediment minor clast, altered by alunite-dickite-silica. (E) Sample D3078029. Phreatomagmatic breccia, polymict, trace sediment - minor clast, altered by alunite-dickite-silica (high grade). (F) Sample D3056884. Crackle sandstone, matrix fill by alunite ± dickite. (G) Sample D3067305. Phreatomagmatic breccia, polymict, altered by alunite-dickite-silica. (H) Sample D3137821. Massive sandstone, oxide staining, pervasive alunite-clay altered. (I) Sample D3035222. Sandstone with alunite vein. (J) Sample D3049860. Phreatomagmatic breccia, altered by alunite-silica, with the alunite matrix partly clast dominant.**



## 3 Geochronology

### 3.1 $^{40}Ar/^{39}Ar$ Geochronology

$^{40}Ar/^{39}Ar$ geochronology is based on the measurement of isotope ratios of a single element, argon. New technology of today has enabled this geochronometer to become one of the most precise systems available to determine the timing and duration of events that accompany mineralisation. Age is determined based on the amount of $^{40}Ar$ produced from radioactive decay of $^{40}K$, so only potassium bearing rocks or minerals can be analysed using this method. Fortunately, potassium is abundant in a range of geological and tectonic settings. One such mineral is alunite, the mineral that is key to this study.

Key locations were determined and ten alunite samples (Figure 2) were chosen: the most pristine alunite zones of each sample were cut out and XRD done for verification of composition and % K-content. These samples were then crushed to 420-240 μm in size and alunite grains were hand-picked and separated under a microscope. These alunite grains have a white-cream colour, soft texture, and anhedral crystal shape with a high potassium content (9-10%). Since the ages were estimated to lie between 1 and 3 Ma, 100-150 mg of alunite per sample were picked with a purity of 99%. Samples were analysed at the argon laboratory in the multi-collector Argus VI at the Australian National University (more information here: http://argon.anu.edu.au/). The samples were step-heated in an ultra-high-vacuum resistance furnace, allowing $^{39}Ar$ diffusion experiments at the same time as $^{40}Ar/^{39}Ar$ geochronology. For each step the gas released is measured in the mass spectrometer, producing one $^{40}Ar/^{39}Ar$ age measurement, corrected for both Cl and Ca contaminants.

Samples were not cleaned using acid ($HNO_3$) to avoid damage, instead being cleansed in deionised water. We assumed that age spectra would have sufficient detail so as to overcome any ambiguity caused by contaminated steps: a major advantage of furnace step-heating methods as opposed to laser analyses. In addition, the methods used involved long periods of cleaning under UHV conditions prior to measurement, thereby maximising the effect of contaminants. Samples were dropped into the resistance furnace and heated to 400°C. This temperature was then immediately reduced, and the sample left for a minimum of 12 hours pumping away unwanted gases. We have been able to show that this simple procedure reduces contamination significantly, especially in the first few steps where low retentivity diffusion domains release their gas and contamination can be abundant.

To calibrate the age information recorded by the mass spectrometer a standard of known age must be irradiated and processed under the same conditions during measurement of the samples from the one irradiation canister. This standard has an independently determined age and is used to determine the fast neutron dose that was received for all samples during irradiation. For the experiments in this study the standard GA1550 biotite (98.5±0.8 Ma) is used (Spell and McDougall, 2003). The reported data have been corrected for system backgrounds, mass discrimination, fluence gradients and atmospheric contamination. Errors associated with the age determinations are one sigma uncertainties and exclude errors in the age of the fluence monitor GA1550. Decay constants used are those of Steiger and Jäger (1977). Each irradiation canister also routinely includes samples that allow accurate calibration of the correction factors used to eliminate interference





from Ca and Cl in producing argon isotopes during the irradiation process. The $^{40}Ar/^{39}Ar$ dating technique is described in detail by McDougall and Harrison (1999) and the methods used here by Forster and Lister (2009).

## 3.2 Age Spectra from Step-heating Experiments

Constraints on timing are critical for understanding regions that have experienced multiple stages of deformation, alteration,
and/or magmatism, such as that found at the Martabe gold field. However, age information can be hard to unravel, especially when there were multiple periods of growth associated with alteration for example, which produce complex apparent age spectrum. Forster and Lister (2004) have addressed this problem and have demonstrated methodology to interpret complex or 'disturbed' age spectra. Their method of 'asymptotes and limits' is used in this research since disturbed age spectra diverge from a single plateau due to mixing of gas from different microstructural reservoirs, commonly occurring as intra-
grain variation, not just from different grains. Using the 'asymptotes and limits' approach allowed the recognition of detailed age information from the pattern of gas released during a single step-heating experiment, including the timing of overprinting growth events.

Age spectra consist of the measured age on the y-axis and the percentage of the cumulated percentage total of the $^{39}Ar$ isotope progressively released during the experiment on the x-axis, calculated retrospectively after the sample has been
completely degassed. Experience showed that there was variation in how alunite grains degas: in this study we demonstrated that alunite samples can degassed quickly over a short temperature range. Therefore, we took care in monitoring the heating schedule for our experiments in order to prevent large amounts of gas from being released in a single step, since such an occurrence can hide mixing and age variation. In contrast, small amounts of gas released with each heating step allows ready identification of different gas populations, with the resultant age spectrum allowing more information to be ascertained from
a single step-heating experiment than any laser spot analysis would provide. Furthermore, the many smaller steps with incremental release of $^{39}Ar$ over a range of different temperatures allows the Arrhenius plot to be uniformly well-populated. This is important when estimating argon diffusion parameters and closure temperatures (Forster *et al*. 2019). We commenced step-heating at 450°C with small increments of 30°C applied until 1000°C was reached, with 50°C steps thereafter.

The age spectra from each of these step-heating experiments are shown in Fig. 3 (a-i). Alunite minerals are known to have
significant contaminants that are released at low temperatures: thus, these samples were heated to 400°C, and then degassed for 12 hours prior to analysis. Even so all of the age plots show evidence of contamination in the first steps, with high initial ages that decrease to a younger plateau age. The experiments could start at a hotter temperature to do away with this contamination, but the release of gas happens at unexpected temperatures and thus the experiments starts where contaminants may still occur. These range from relatively small amounts of initial gas released (0-2%) for samples
PURNAMA P-01 and D3137821, to larger amounts (10-20%) for the remaining samples. Furthermore, during the last steps of each of the heating schedule there is also contamination present. These final steps have produced large error bars and also trend towards a younger age. We have excluded steps with such contamination during age selection.

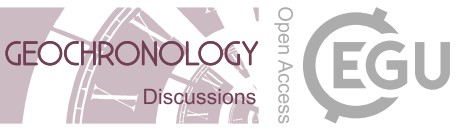

Figure panels (a)–(j): Argon release spectra (Apparent Age (Ma) ± 2σ vs. Percentage ³⁹Ar released) for ANU CAN #30 alunite samples.

(a) ANU CAN #30, Name: PURNAMA P-01, Foil: A1, Mineral: Alunite, Mass: 85.9mg, Steps: 35; Age 2.21 ± 0.02 Ma (95% c.l.) MSWD = 0.99; Age 1.99 ± 0.03 Ma (95% c.l.) MSWD = 0.59

(b) ANU CAN #30, Name: D3011643, Foil: A2, Mineral: Alunite, Mass: 130.4mg, Steps: 30; Age 3.47 ± 0.01 Ma (95% c.l.) MSWD = 0.03; Age 3.23 ± 0.01 Ma (95% c.l.) MSWD = 0.02

(c) ANU CAN #30, Name: D3150595, Foil: A3, Mineral: Alunite, Mass: 143.7mg, Steps: 32; Age 2.42 ± 0.09 Ma (95% c.l.) MSWD = 1.29

(d) ANU CAN #30, Name: D3112423, Foil: A4, Mineral: Alunite, Mass: 149mg, Steps: 32; Age 1.99 ± 0.07 Ma (95% c.l.) MSWD = 1.77

(e) ANU CAN #30, Name: D3078029, Foil: A5, Mineral: Alunite, Mass: 143.1mg, Steps: 32; Age 2.32 ± 0.14 Ma (95% c.l.) MSWD = 1.95

(f) ANU CAN #30, Name: D3056884, Foil: A6, Mineral: Alunite, Mass: 138.5mg, Steps: 32; Age 2.18 ± 0.02 Ma (95% c.l.) MSWD = 0.30; Age 1.94 ± 0.04 Ma (95% c.l.) MSWD = 0.23

(g) ANU CAN #30, Name: D3067305, Foil: A7, Mineral: Alunite, Mass: 124.7mg, Steps: 32; Age 2.22 ± 0.10 Ma (95% c.l.) MSWD = 0.47

(h) ANU CAN #30, Name: D3137821, Foil: A8, Mineral: Alunite, Mass: 143.8mg, Steps: 32; Age 2.06 ± 0.02 Ma (95% c.l.) MSWD = 0.35; Age 1.47 ± 0.07 Ma (95% c.l.) MSWD = 0.58

(i) ANU CAN #30, Name: D3035222, Foil: A9, Mineral: Alunite, Mass: 122.9mg, Steps: 32; Age 2.27 ± 0.03 Ma (95% c.l.) MSWD = 0.92

(j) ANU CAN #30, Name: D3049860, Foil: A10, Mineral: Alunite, Mass: 131.2mg, Steps: 32; Age 2.03 ± 0.04 Ma (95% c.l.) MSWD = 1.25; Age 1.55 ± 0.15 Ma (95% c.l.)






**Figure 3. a-j. Age data from step-heating experiments. Plots created using the *eArgon* program. Selected steps used for age data are shown in either green or blue. Errors on each step are shown as two sigma uncertainties.**

### 3.3 Probability Curve Analysis

To determine the periods of alunite growth each data point was plotted as a Gaussian distribution (Fig. 4). Five distinct
periods of alunite growth could thereby be distinguished across the 15 ages measured in this study. The oldest event recorded (Event 1) occurs between 3.48–3.46 Ma. The second oldest event (Event 2) occurs between 3.24–3.22 Ma. Alunite is then recorded growing at 2.51 to 2.12 Ma (Event 3) and between 2.08 and 1.90 Ma (Event 4). The youngest growth is found to be from 1.70 to 1.40 Ma (Event 5). However, to understand what these ages mean, it is necessary to evaluate the diffusion properties for alunite, to estimate closure temperatures.

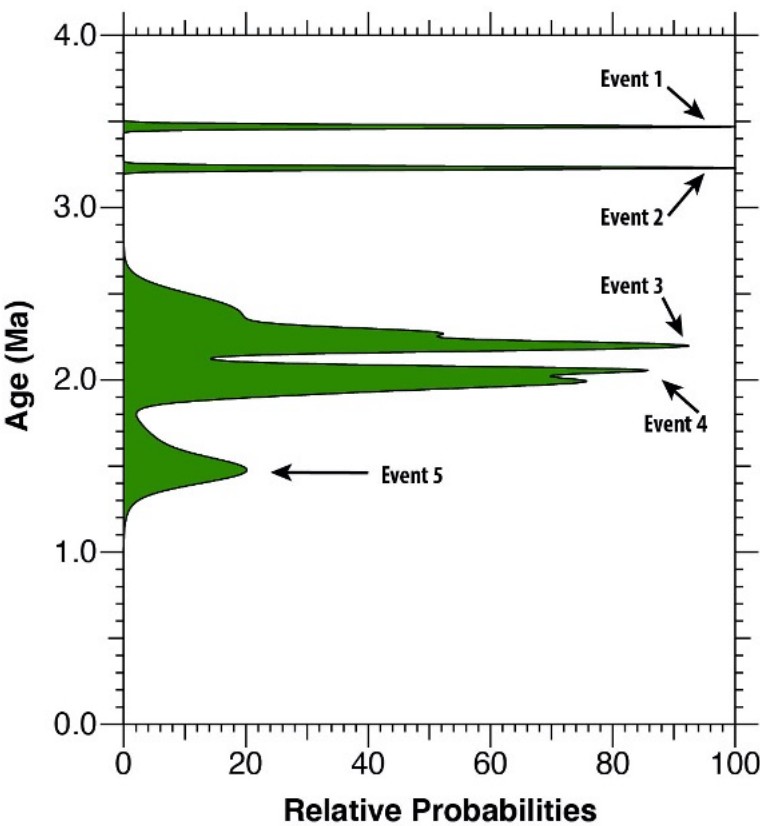


**Figure 4. Gaussian distribution curve of the $^{40}$Ar/$^{39}$Ar ages. The five alunite growth events are labelled from oldest to youngest.**

### 4 Arrhenius data and closure temperature estimates

An age spectrum reflects either the age at which a mineral or domain grew or was reset by grain boundary migration, or when the mineral system cooled below its closure temperature. For rapid cooling, the closure temperature marks the point at



which diffusion of a daughter product out of the mineral system becomes sufficiently slow as to be safely ignored. The daughter isotope, in this case $^{40}$Ar, thereafter accumulates in the mineral lattice. However, the closure temperature is not a single well-defined temperature, but a complex range which can differ between domains even within a single crystal. Estimates for closure temperature are thus important to understand the context of age data. Since epithermal deposits form at temperatures ranging from 200-350°C (White and Hedenquist, 1990), retentive alunite could yield growth ages. Conversely,

if alunite is unretentive, the age would indicate when the deposit cooled, and alunite could not be able to be used to distinguish different overprinting fluid systems.

Closure temperature can be determined from Arrhenius plots as outlined by Dodson (1973). Arrhenius plots take the inverse of absolute temperature from each step of the heating schedule along the x-axis and logarithm of $D_0/r^2$, where $D_0$ is diffusivity and r is the radius of the diffusion domain, on the y-axis (McDougall, 1999). According to Lovera *et al.* (1997)

the Arrhenius plot should display a single straight line, when dealing with a single diffusion domain. From this line we can determine the diffusion paraments (*i.e.,* activation energy and frequency factor) and calculate closure temperature by inferring diffusion geometry and assuming cooling rates were faster than 20°C per million years.

There has been much debate around interpretation of Arrhenius data. Forster and Lister (2010) simulated artificial step-heating experiments in order to determine a consistent methodology for interpreting Arrhenius plots, especially those that do

not produce linear arrays. From this work came the Fundamental Asymmetry Principle (FAP) which outlines objective guidelines for the analysis of Arrhenius plots. Simply put, these guidelines follow that any line must divide the sequence of steps by rank order. Points of a higher rank order will lie on or to the left of the line and points with a lower rank order with lie on or to the right of the line. The FAP and the need to well populate the Arrhenius plots particularly in the lower temperature of the experiment is absolutely essential to gaining closure temperature data and to allow modelling.

Using these methods, we analysed the ten Arrhenius plots from our alunite step-heating experiments, using the *eArgon* software (Fig. 5). By obeying the FAP it was found that the activation energy ranges between 360 – 500 kcal/mol, with frequency factor ranging between 1.89e$^{14}$s$^{-1}$ – 8.62e$^{18}$s$^{-1}$. Closure temperature estimates range between 390°C and 519°C for cooling rates of 20°C/Ma. Therefore, given the temperatures involved in epithermal mineralisation, these alunite samples were able to accurately determine the age of crystallisation in the various fluid systems involved.





Figure 5. a-j. Arrhenius data for the step-heating diffusion experiments. Plots were created using the *eArgon* program. Lines have been fitted according to the Fundamental Asymmetry Principle (Forster and Lister, 2010). Steps with less than 0.02% gas released were excluded.




## 5 Discussion

$^{40}Ar/^{39}Ar$ geochronology has proven to be essential to timing the growth and alteration of minerals in the Martabe gold mine. The first research recognising this was Sutopo (2013) who analysed four alunite samples from the Martabe deposits using $^{40}Ar/^{39}Ar$ geochronology. Two of the experiments used the step-heating technique, while the other two used the total fusion age. This research concluded that alunite grew between 3.30 Ma and 2.0 Ma. Our $^{40}Ar/^{39}Ar$ geochronology results for ten samples (Fig. 3) extended the range of the activity in this alteration system to be between 1.40 Ma and 3.48 Ma. Five of the

measured samples (Figs. 3 a, b, f, h, j) display complex age spectra, and a single plateau is not present. These involve mixing between radiogenic argon released from alunite grown at different times. Such spectra can be interpreted according to the method of 'asymptotes and limits' (Forster and Lister, 2004), in consequence unveiling the influence of two overprinting alunite forming events in the same age spectrum. This is the first example of overprinting events being recorded in a single step-heating experiment of alunite.

Complexity increases when the interpreted ages across nine of the samples show periods of alunite growth at 1.4 to 1.7 Ma, 1.9 to 2.1 Ma, 2.1 to 2.5 Ma.  In contrast, sample D3011643 produced older events at $3.47 \pm 0.01$ Ma and $3.23 \pm 0.01$ Ma, which overlap in timing with the emplacement of the dacite $3.8 \pm 0.5$ Ma (dated using the U-Pb method) and andesite $3.1 \pm 0.4$ to $2.8 \pm 0.3$ Ma (also dated using the U-Pb method) flow dome complexes (Sutopo, 2013). The clustering of ages and the overlap with flow-dome formation gives us confidence that the interpretation of age spectra using the method of 'asymptotes

and limits' is in fact real and not simply an artefact of contamination or excess argon. In total there have been five separate alunite forming periods recorded in our $^{40}Ar/39Ar$ step-heating experiments. The younger periods of alteration could be driven by the activity of the Sumatran Fault System, which reactivated fluid pathways. The interpreted $^{40}Ar/^{39}Ar$ ages are shown in Figure 6, in context of their 2D sample location. More geological information is needed to constrain the alunite alteration systems further, this is difficult due to the area being covered by dense rain forest and restricted zones.





**Figure 6. Map of Martabe gold field with alteration systems mapped. Alteration systems outlined according to the sample location and period of alunite growth. The two older events are grouped together.**

This study has also shed light on the closure temperature of alunite. There have been previous estimates for the closure temperature of alunite, which have ranged from ~240-320°C (Love et al., 1998, Arribas et al., 2011). This is within the temperature range in which epithermal deposits are known to form, *i.e.*, 200-350°C (White and Hedenquist, 1990). If alunite was so unretentive of radiogenic argon, the ages would represent when the epithermal deposit cooled below the closure temperature rather than when it had formed. It is important, therefore, that our results show otherwise, unleashing the potential for alunite geochronology to map different fluid systems formed in a narrow time range. Key to this effort was the






rigorous testing involved in ultra-high-vacuum (UHV) [39]Ar diffusion experiments with step-heating schedules designed to

allow sufficient detail and spatial coverage in the Arrhenus plot to allow application of the Fundamental Asymmetry Principle (or FAP). Our diffusion experiments produced Arrhenius data that imply closure temperatures between 390°C and 519°C for modest cooling rates (Fig. 5). This indicates that alunite can be highly retentive of argon and thus can be used as a reliable mineral for [40]Ar/[39]Ar geochronology timing ore deposition in epithermal systems. Radiogenic argon was retained even during later overprinting hydrothermal or magmatic events such as those found in this study.

Similarly retentive results were obtained by Ren and Vasconcelos (2019) for some of their samples, although these authors did not apply the FAP. Cursory examination of the Arrhenius plots reveals several examples in which these authors thereby underestimated the activation energy and associated retentivity.

**6 Conclusion**

Through a detailed and suitable heating schedule, we were able to degas alunite samples efficiently, without losing a large
percentage of gas in a single step. This allowed us to extract maximum amount of information from both the age spectra and the Arrhenius plots. Where applicable the age spectra were interpreted using the method of asymptotes and limits, allowing distinction of overprinting fluid systems across five of the samples. These new [40]Ar/[39]Ar ages indicate that there were five peak periods of alunite growth around the Martabe deposits at 1.40–1.70 Ma, 1.90–2.08 Ma, 2.12–2.51 Ma and 3.22 and 3.48 Ma. This extends the known duration of acid-sulphate alteration that had taken place at Martabe, with distinct periods of
fluid activity that could help future mineral exploration within the Martabe gold field.

Analysis of the Arrhenius plots put the closure temperature of alunite ranging between 390°C and 519°C, which is above the temperature expected for the formation of the Martabe deposits. This result, and the heterogeneity in age, gives confidence that the measured ages from [40]Ar/[39]Ar geochronology are for the formation of alunite and that these are not cooling ages.

Although the map of the extent of the overprinting fluid systems is not complete, and lacks detail, it is evident that
'vectoring' using mineral compositions could be fraught, unless samples were taken from the same fluid system. It is also evident that mineralisation in the Purnama pit is the result of two specific fluid events, one enriching the other.

**Data availability**

All data collected in this study and presented in this article are provided in the MPhil thesis of Jack Muston submitted to the Research School of Earth Sciences (RSES), Australian National University.

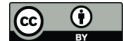



**Author contribution**

All authors contributed to the discussion and reviewing of the manuscript. The paper includes part of a MPhil thesis by the first author, supervised by Prof. Gordon Lister.

**Disclaimer**

This article includes part of the MPhil Thesis of the first author submitted to the Research School of Earth Sciences (RSES), Australian National University.

**Conflicts of Interest**

The authors declare that they have no conflicts of interest.

**Funding Statement**

Research has been funded by the Australian Research Council Linkage Project LP130100134 – "Where to find giant porphyry and epithermal gold and copper deposits" LP130100134 "Where to find giant porphyry and epithermal gold and copper deposits"]. Sample Irradiations were paid by PT Agincourt.

**Acknowledgments**

Shane Paxton and Sareh Rajabi are thanked for their assistance with mineral separations, and Davood Vasegh for his work in sample processing and analysis at the RSES Argon Laboratory. The team at PT Agincourt is thanked for providing the alunite samples.

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
