# Peer review of "Direct dating of overprinting fluid systems in the Martabe epithermal gold deposit using highly retentive alunite"

_Geochronology, 2020_

## Referee Comment (RC1) · Anonymous Referee #1 · 14 Jan 2021

Muston et al present 40Ar/39Ar results from alunite in ten samples from the Martabe gold deposit. They also obtained closure temperature constraints from the alunite which ranged from 390-519C. Based on their results and interpretations, they claim that the gold deposits were formed in 5 discrete growth events. Overall, the text is well written, but it does read like a 40Ar/39Ar textbook in several places. The figures are clear.

The major flaw of this manuscript is that the authors have overinterpreted the age spectra that have been generated and thus one section (section 3.3), two figures (Figures 4 and 6), and the conclusions of the paper (5 discrete pulse events) are not supported by

these data. As noted by the authors, the age spectrum from almost all of the samples are complicated. Numerous criteria have been put forth over the last several decades to evaluate age spectra and to calculate a plateau ages (e.g., Fleck et al., 1977; Sharp and Renne, 2005; Jourdan et al., 2004). In a recent paper consisting of forty 40Ar/39Ar specialists from around the globe, the following criteria were put forth for a plateau: (1) consist of at least five or more consecutive steps that comprise at least >50% of the 39Ar released; (2) not have a slope (i.e., the majority of consecutive plateau steps do not have ascending or descending ages; Sharp and Renne, 2005); and (3) have an isochron regressed through all of the plateau steps with a (40Ar/36Ar)i that is indistinguishable from the atmospheric value at the 95% confidence level. Almost all of the Muston et al spectra have a significant slope. Only one sample (D3035222) has a plateau based on the Schaen et al (2020) criteria. Moreover, there are no accompanying isochrons with the spectra so the 40Ar/36Ar intercepts cannot be evaluated. Muston et al have selected random concordant steps in many samples to be "plateaus". This is gross overinterpretation. There is not strong evidence for 5 pulses of mineralization and there is no evidence to support the claim that these new data extend the range of alteration beyond what was defined by Sutopo (2013) from 3.3 to 2.0 Ma. The authors claim that the complicated spectra involve mixing between radiogenic argon released from alunite grown at different times based on the method of 'asymptotes and limits' (Forster and Lister, 2004). There is simply not enough data to support this claim. In sum, this manuscript has major flaws and does not warrant publication in GChron in its current form.

Other editorial and interpretive comments: Figure 1: caption says there are 10 samples. Only 9 are shown. Purnama P-1 needs to be added to this figure as it is in figure 6.

Lines 46-47: The host rocks are a series. . .. The text lists several kinds of hosts

Lines 80-84: The furnace experiments were done so that the diffusion experiments could be done simultaneously. This is the advantage of the furnace analyses compared

to a laser unless a laser is equipped with a calibrated optical pyrometer. However, on line 81, the authors claim that furnace experiments have a "major advantage" over laser experiments in that they can produce age spectra with sufficient detail. This statement is misleading and needs to be omitted. Laser analyses of WAY less material than what was used here would also yield age spectra with sufficient detail (see Pan et al., 2019 Economic Geology or Holm et al 2019).

Line 84: the extensive cleaning should minimize the effect of contaminants. Text says maximize.

Line 85: the text is written as if they have used a novel furnace technique. People have been dropping samples into a furnace and degassing them prior to 40Ar/39Ar incremental heating experiments since the 1970's. See Staudacher et al 1978 or Mc-Dougall and Harrison, 1988. The text repeatedly states how they did detailed furnace incremental heating experiments in many small steps. However, it doesn't say how the furnace blanks were done. Were they also done in many small increments or did they do a few and interpolate in between furnace blank temperatures? Almost all of these experiments yielded spectra with a similar shape which points towards the spectra being a function of the experiments and not the samples all behaving the same way. Note that the shape of their alunite spectra look very different than the relatively flat spectra generated in a recent study of alunite and jarosite by Ren and Vasconcelos (2019).

Line 121: the authors keep using the word contaminants. The older apparent ages at the beginning of the experiments are stated to be "evidence of contamination." More descriptive detail is needed. The older ages could be due to low temperature excess Ar or it could be attributed to 39Ar recoil (see Jourdan and Renne, 2013). They also say that the steps at the end of the experiment are due to contamination. This could also be due to recoil. Given that Muston et al did not evaluate 39Ar recoil, it should be listed as a possibility in both cases. Accompanying K/Ca or K/Cl plots should be included so that one could evaluate how these variables evolve as the apparent ages change in the samples.

---

## Referee Comment (RC2) · Anonymous Referee #2 · 17 Feb 2021

Overview :

Muston et al. present an interesting data set and pose an important question about the recurrent nature of fluid flow events in an economic deposit. Insufficient petrologic information is given about the analysed materials, hampering interpretation of the data. The argon data treatment is difficult to justify as it is both incomplete (no inverse isochrons given, etc.) and ignores more straightforward explanations, e.g., simple, robust plateau definitions. Indeed, the selection of steps appears arbitrary in many cases. Overall, the manuscript is under-developed and the given interpretations are unconvincing, and for this reason I would reject this paper for publication in GChron.

[Figure]

General Comments :

Line 80 : No detailed chemical, mineralogic or imaging data are given for the samples, so the presence of silicate minerals cannot be evaluated at a scale relevant to the material prepared for analysis. Do we know that this material is exclusively alunite or even dominantly alunite?

Line 81 : How do furnace spectra discriminate contaminants whereas lasers do not? This statement is incorrect and unsupported. Lasers can step-heat and some laser systems are equipped with pyrometers. Given then that blanks can be measured throughout a laser step-heating sequence and that furnaces have comparatively long heat-up and cool-down times, many would argue that laser step heating is superior to furnace step heating. If you're implying that lasers are only used for 'spot analysis', i.e., total fusion, then the same can be done with a furnace.

Line 83 : Clarify what is meant by 'contaminants'. Do you mean silicate contaminants? Excess argon? Atmospheric argon? This statement seems contradictory – why would you employ a method that maximises the effect of contaminants?

Line 84 : How long is the pre-degas step at 400°C? Is age information lost here? Recoil effects? If gas is lost from low-retentivity sites, how does this affect the 'asymptotes & limits' (AL) data interpretation?

Line 120 : Again – contaminants of what? Wouldn't you expect silicate contaminants (micas or relict feldspar) to degas at higher temperatures than alunite?

Line 121 : Why is this not a recoil effect? What about fluid inclusions with excess argon?

Line 126 : what is the nature of contamination that produces younger ages? Is it a mineral? Clays?

Line 134 : What does 'each data point' refer to? Inferred ages for the selected plateaux or the calculated ages of each gas aliquot from all experiments?

Line 190 : This argument would be better supported if inverse isochrons of the included steps for the AL treatment were shown, i.e., demonstrating that excess Ar, etc., is not an issue.

Figure 3.

Need inverse isochrons to examine presence or absence or excess argon, etc.

Need to list step selection criteria for plateau determination – many cases where steps are left out with no obvious reason for their exclusion.

Plots would benefit from showing %radiogenic gas and Cl/K, Ca/K per step.

A: Are the green steps subject to recoil or excess 40Ar? Why is the ca. 90% step left out of the blue plateau? B: What is the justification for subdividing this into two 2-step plateaux rather than one 7 step plateau? C: Seems an arbitrary choice of 3 steps – five other 'sets of three' could be chosen, or a large set of five steps (the final five). Etc..

Technical Points :

Figure 1 : Sample map shows only 9 sample labels (not 10) and 8 locations. PUR-NAMA appears to be missing. For the location with two samples, what is the spatial relationship of these two?

Line 191 : superscript 39Ar

Line 111 : can 'be' degassed. . .

―――――――――――――――――――――――――